# Inexpensive Screening Method to Validate the Efficacy of Ethanedinitrile Fumigant on the Forest Invasive Nematode Pest *Bursaphelenchus xylophilus*

**Ondřej Douda [1,*], Václav Stejskal [1,2], Marie Manasova [2], Miloslav Zouhar [2] and Jonáš Hnatek [2,3]**

[1]  Division of Plant Health, Crop Research Institute Prague, Drnovská 507,
    161 06 Prague 6, Ruzyně, Czech Republic; stejskal@vurv.cz
[2]  Department of Plant Protection, Faculty of Agrobiology Food and Natural Resources,
    Czech University of Life Sciences Prague, Kamýcká 129, 165 00 Prague 6, Suchdol, Czech Republic;
    manasova@af.czu.cz (M.M.); zouhar@af.czu.cz (M.Z.); jonas.hnatek@draslovka.cz (J.H.)
[3]  Lučební závody Draslovka a.s., Havlíčkova 605, 280 99 Kolín IV, Czech Republic
*   Correspondence: douda@vurv.cz

**Abstract:** At a global scale, the sustainability of forests is endangered by multiple invasive species, including the pine wood nematode (*Bursaphelenchus xylophilus)*, a quarantine pest. International laws and standards require that all exported wood coming from countries in which *B. xylophilus* is present be chemically or physically treated. Since a major fumigant, methyl bromide, was banned, there has been a need to generate data for alternative fumigants, such as ethanedinitrile (EDN), for this purpose. Since the field screening of fumigants (i.e., the application of various doses to and exposure times of naturally infested wood logs) is prohibitively expensive, the aim of this study was to develop a quick and inexpensive laboratory method. Here, we suggest and describe an innovative method based on sawdust cultures for EDN efficacy screening. In the validation part of this study, we demonstrated (i) the high survival of the nematodes in the sawdust and (ii) the high efficacy of EDN against this pest under in vitro conditions; 100% mortality was observed after 6 h of EDN exposure to a dose of $25 \text{ g/m}^3$. In particular, our newly developed model system could be used for the initial screening of various doses of and exposure protocols for EDN and similar types of fumigants developed with the intention of regulating *B. xylophilus* occurrence in exported wood. It is believed that the validated method may help to develop new and effective EDN fumigation procedures and thereby contribute to the long-term protection of forests worldwide.

**Keywords:** *Bursaphelenchus xylophilus*; ethanedinitrile; EDN; fumigation; model testing

## 1. Introduction

At a global scale, the sustainability of forests is endangered by multiple invasive species [1]. The prevention of the spread and establishment of any quarantine pest organism requires the availability of cost-effective and rapid phytoquarantine measures [2]. However, since the ban of the major fumigant methyl bromide, rapid and equally efficient chemical methods for the control of certain quarantine pests are still lacking. The pine wood nematode (*Bursaphelenchus xylophilus*; (Steiner and Buhrer) Nickle, Nematoda), a serious pest of European and Asian *Pinus* conifers, is not an exception. Although the outbreak of this phytoparasitic nematode in Portugal [3] led to the adoption of immense control measures, including the establishment of a clear cut zone [4], the current situation, with a recent *B. xylophilus* detection in Spain [5] and common occurrence of *B. xylophilus* beetle vectors (*Monochamus* sp.) in Europe, provides a chance for this pest to further expand on this

continent [6]. Moreover, recent modeling suggests the possibility of *B. xylophilus* establishment in central Europe [7]. The presence of *B. xylophilus* in Europe presents a threat to sustainable forest management and natural balance over the entire continent. For this reason, all wood imported from countries where *B. xylophilus* is native (i.e., North America) or has been introduced (i.e., Japan, China, Korea and Europe) must be treated to kill any nematode present within the wood tissues [8]. Since there is no registered fumigant, the only currently available method is heat treatment [8]. However, this method has several drawbacks. The main disadvantages of heat treatment include unwanted changes in the wood mechanical properties after heat exposure, the cost of energy, and the construction and logistic costs associated with the need for special heat treatment chambers and wood transportation from forests. Therefore, an effort to develop an alternative method utilizing chemical treatments exists. Research on this topic intensified after methyl bromide was banned [9], as this substance was routinely used as a wood fumigant. The spectrum of the tested active ingredients includes sulfuryl fluoride [10], hydrogen cyanide [11,12] and ethanedinitrile (EDN) [13,14]. EDN, in particular, has certain advantages when used as a wood fumigant: it penetrates wood fairly well and is heavier than air and other gasses; therefore, it is easy to apply and seal under a sheet (e.g., on lumber stored in stockpiles covered with plastic sheets). In addition, it decomposes rapidly into nontoxic products [15] and does not interact with the treated wood. As with other toxic fumigants, EDN has to be applied by trained personnel. The full registration of EDN for wood fumigation in the European Union is currently in progress. However, temporary registration has been issued [16] to mitigate the European spruce bark beetle (*Ips typographus*, Linnaeus) disaster in the Czech Republic. To date, more than 70,000 m$^3$ of wood logs infested with bark beetles have been fumigated using EDN. The efficacy of EDN against insect pests was also confirmed with the burnt pine longhorn beetle (*Arhopalus ferus* (Mulsant)) [17]. EDN has potential as a phytosanitary alternative to methyl bromide (MB) for the treatment of logs [18].

The efficacy of fumigants is affected by environmental conditions. The crucial conditions consist of humidity and temperature. In particular, the latter factor affects the velocity of invertebrate pest population development, as well as pest metabolism and breathing rate [19]. The high efficacy of EDN on the mortality of *B. xylophilus* individuals was confirmed previously in field trials under varying physical conditions [13,14]. These trials were conducted with the use of naturally infested logs, which were treated and covered by polyethylene sheets, and naturally infested *Pinus* branches, which were placed in glass desiccators [14].

The main aim of this study was to develop a quick and inexpensive laboratory method (based on sawdust cultures) for testing EDN efficacy, which will have certain advantages over the methods used until now. This method makes it possible to test EDN efficacy under different temperatures and moisture conditions, and naturally infested wood is not needed, so screening could also be done in countries where *B. xylophilus* is not yet present. Furthermore, it is possible to standardize this newly developed method, e.g., as a component of a standard operating procedure for EDN testing under various environmental conditions.

## 2. Material and Methods

The tests were conducted in 50 mL polyethylene plastic tubes covered by nylon fabric (mesh size of 25 μm) to allow the free movement of the fumigant. Fresh Scots pine (*Pinus sylvestris*, Linnaeus) was obtained from the forest in the Central Bohemia region (GPS coordinates: 50.1439014N, 14.3191069E), and the lack of *Bursaphelenchus* sp. nematodes in the pine was confirmed by Baermann's funnel method. Sawdust was created from pine logs using a chainsaw and stored at −20 °C prior to the experiment. Twenty grams of sawdust was added to each tube; 1 mL of nematode suspension containing all the developmental stages obtained from *B. xylophilus* individuals maintained on *Botrytis cinerea* (De Bary, Whetzel) was pipetted into the sawdust. One milliliter contained between approx. 300 and 800 *B. xylophilus* specimens; this was established by manually counting the samples under a stereomicroscope. The average resulting moisture content of the treated sawdust was 39%; this value was obtained by weighing the sawdust prepared for the experiment, drying it at 105 °C for 24 h,

and weighing the dried sawdust. After preparation, the plastic tubes were placed into a stainless steel fumigation chamber (see details in Stejskal et al., 2014) and treated with EDN (99% purity, supplied by Lučební závody Draslovka, Kolín, Czech Republic) at concentrations of 25 and 50 gm$^3$. The following exposure times were tested: 6, 12, 18, 24, 30 and 40 h. The temperature in the fumigation chamber was 22 ± 3 °C, and the choice of temperature and moisture was based on the results of our previous tests with hydrogen cyanide [12]. Five tubes were treated during each application, and the application was repeated three times. The samples of the atmosphere in the chambers were withdrawn during fumigation using a rubber septum and plastic syringe in the designated intervals, and the EDN concentration was established using gas chromatography (Shimadzu GC−17A, RT-QPLOT, 30 m, ID 0.53 mm, Software Clarity DataApex; Shimadzu Corp., Kyoto, Japan). The gas chromatography method was based on comparing the detector response from a sample with the response from an external standard with a known concentration. After fumigation, the nematodes were extracted from the samples using a modified Baermann's funnel method, enabling the extraction of the nematodes directly from the plastic tubes without the need to remove the sawdust (utility model number CZ 24,090, Figures 1 and 2), and the collected nematodes were counted under a stereomicroscope after 12 h of extraction. The untreated control was evaluated in the same manner after 40 h of storage beside the fumigation chamber and 12 h of extraction. The obtained data were summarized using the mean and standard error of the mean (Statistica 13.3 software; Dell Software, Inc., Round Rock, TX, USA), and the CT product (concentration x time) values were established as described by [20].

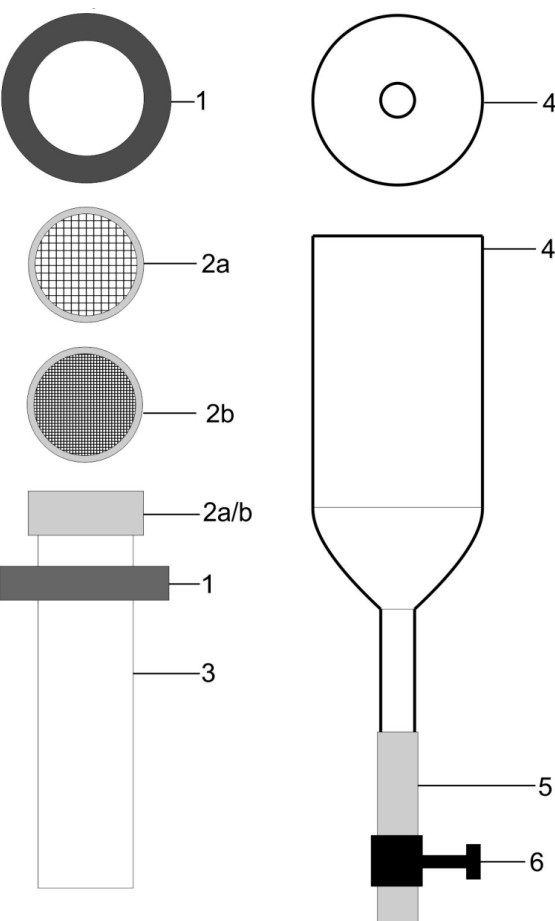

**Figure 1.** Scheme of utility model no. CZ 24090 U1. Legend: 1—rubber seal, 2a—grid cap for the extraction of nematodes, 2b—grid cap for fumigation, 3—50 mL falcon tube, 4—glass funnel, 5—silicone tube, and 6—tube clip.

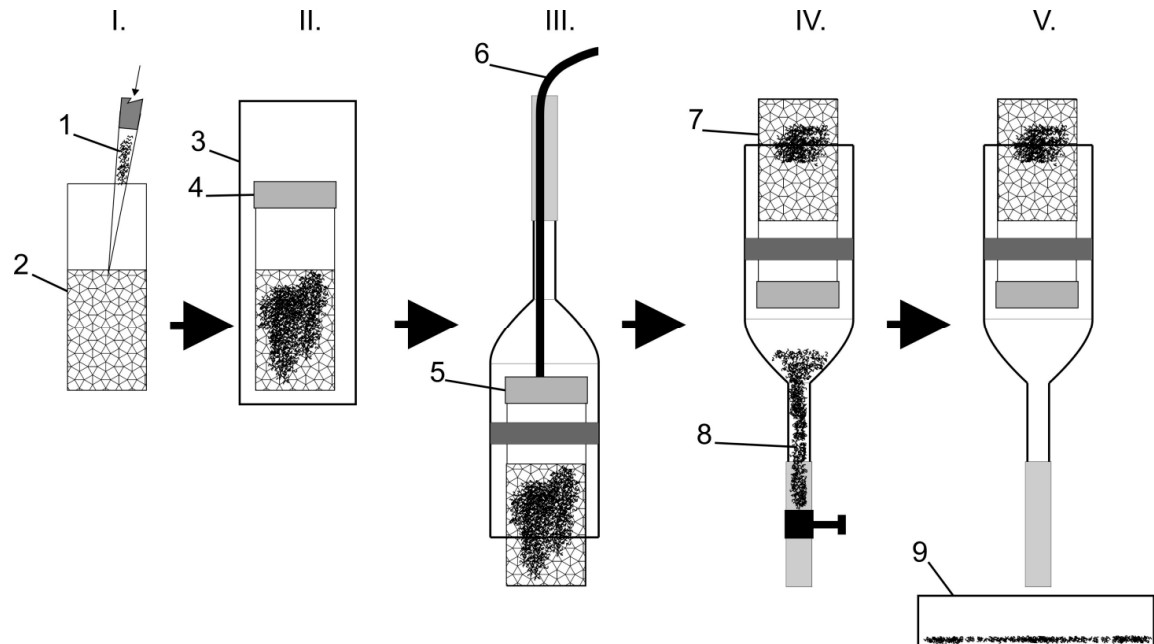

**Figure 2.** Scheme of the ethanedinitrile (EDN) efficacy screening process. Legend: I.—inoculation of the nematode suspension into sawdust, II.—fumigation, III.—filling the system with water using a filling tube, IV.—extraction of the surviving (moving) nematodes from the sawdust, and V.—counting the surviving nematodes; 1—nematodes, 2—sawdust, 3—fumigation chamber, 4—grid cap for fumigation (Figure 1(2b)), 5—grid cap for extraction (Figure 1(2a)), 6—filling tube, 7—dead nematodes, 8—surviving nematodes, 9—counting dish with the surviving nematodes.

## 3. Results

Our work confirmed sawdust as a suitable medium since, on average, from 302 to 837 nematode specimens were extracted from the untreated control tubes, even after 40 h (Tables 1 and 2). The moisture content of sawdust is the key factor in nematode survival, and the 39% moisture content used in our experimental design is sufficient to keep the nematodes alive for this time. One hundred percent nematode mortality was observed after 6 h of treatment at the lowest tested ethanedinitrile concentration (25 $g/m^3$, Table 1 and 50 $g/m^3$ Table 2). The CT product values were established for both of the tested initial concentrations; the average CT products for the initial doses of 25 and 50 $gm^3$ were lower than 157.34 and 311.57 $g*h/m^3$, respectively (Tables 3 and 4), as no surviving nematodes were observed in any treatment.

**Table 1.** Number of living nematodes after the ethanedinitrile treatment with an initial dose of 25 $g/m^3$ (average ± standard error).

| Replication | n | Untreated Control | 6 h | 12 h | 18 h | 24 h | 30 h | 40 h |
|---|---|---|---|---|---|---|---|---|
| 1 | 5 | 432.00 ± 59.76 | 0.00 ± 0.00 | 0.00 ± 0.00 | 0.00 ± 0.00 | 0.00 ± 0.00 | 0.00 ± 0.00 | 0.00 ± 0.00 |
| 2 | 5 | 327.60 ± 17.47 | 0.00 ± 0.00 | 0.00 ± 0.00 | 0.00 ± 0.00 | 0.00 ± 0.00 | 0.00 ± 0.00 | 0.00 ± 0.00 |
| 3 | 5 | 302.00 ± 16.54 | 0.00 ± 0.00 | 0.00 ± 0.00 | 0.00 ± 0.00 | 0.00 ± 0.00 | 0.00 ± 0.00 | 0.00 ± 0.00 |

**Table 2.** Number of living nematodes after the ethanedinitrile treatment with an initial dose of 50 $gm^3$ (average ± standard error).

| Replication | n | Untreated Control | 6 h | 12 h | 18 h | 24 h | 30 h | 40 h |
|---|---|---|---|---|---|---|---|---|
| 1 | 5 | 564.20 ± 51.05 | 0.00 ± 0.00 | 0.00 ± 0.00 | 0.00 ± 0.00 | 0.00 ± 0.00 | 0.00 ± 0.00 | 0.00 ± 0.00 |
| 2 | 5 | 837.00 ± 48.70 | 0.00 ± 0.00 | 0.00 ± 0.00 | 0.00 ± 0.00 | 0.00 ± 0.00 | 0.00 ± 0.00 | 0.00 ± 0.00 |
| 3 | 5 | 595.00 ± 49.04 | 0.00 ± 0.00 | 0.00 ± 0.00 | 0.00 ± 0.00 | 0.00 ± 0.00 | 0.00 ± 0.00 | 0.00 ± 0.00 |

**Table 3.** CT products for an initial dose of 25 g/m$^3$.

| Replication | CT Product (g*h/m$^3$) |
|---|---|
| 1 | 150.13 |
| 2 | 166.36 |
| 3 | 155.52 |
| Average ± standard error | 157.34 ± 4.77 |

**Table 4.** CT products for an initial dose of 50 g/m$^3$.

| Replication | CT Product (g*h/m$^3$) |
|---|---|
| 1 | 296.25 |
| 2 | 313.83 |
| 3 | 324.64 |
| Average ± standard error | 311.57 ± 8.27 |

## 4. Discussion

An evaluation of the fumigation effect of EDN on *B. xylophilus* present in naturally infested logs has already been performed [13,14], with promising results. The suggested innovative method is based on the use of nematode cultures in sawdust sealed with nylon mesh instead of wooden logs naturally infested with nematodes. It is impossible to perform research on the efficacy of ethanedinitrile against *B. xylophilus* in naturally infested logs under conditions found in the Czech Republic in a fumigation chamber, as the pest is not present in this country and the transport of infested wood would present an unacceptable phytosanitary risk. In this situation, the described method is exploitable for the evaluation of the effect of fumigation on nematode mortality as well as for model fumigations under different temperature conditions at different moisture levels and EDN concentrations. The density of nematodes in the wood may also be a factor conditioning the efficacy of the fumigants. The numbers of nematodes in our untreated controls fluctuated from approximately 300 to 800 specimens; however, it seems that the overall results obtained from the treated variants were not affected by differences in the density of nematode inoculum. Originally, this experimental design was used for research on hydrogen cyanide [12], a substance with considerably different physical and chemical properties (density and level of sorption into the wood) than those of EDN, but comparable results were obtained to those of the test with EDN. Our screening showed that testing fumigation on nematodes present in sawdust is a functional method that could be valuable, especially for initial trials conducted with new fumigants. Our method of testing is relatively simple and can be used instantly in practical applications. The substantial advantage of the tested method is that it is easily applicable to various substances and produces fully comparable results, which is not the case for naturally infested wood. On the other hand, we observed 100% mortality with considerably lower EDN concentrations than those used in tests with naturally infested logs [14]; this could be caused by the higher inner surface area of the material exposed to the fumigant in our tests. The surface area of a log is the most important factor influencing fumigant sorption and desorption rates, with a greater surface area resulting in greater (de)sorption rates, as was shown in the case of methyl bromide [21]. We can expect that higher fumigant concentrations and exposure times will be needed to achieve similar results in naturally infested samples. This seems to also be the case for the results obtained by [13], who previously achieved 98% mortality with an EDN concentration of only 158 g/m$^3$ and exposure lasting for 72 h.

## 5. Conclusions

In our opinion, testing with sawdust could be particularly useful for the preliminary screening of the activity of new fumigants against *B. xylophilus* nematodes. Although this method cannot fully substitute field trials in naturally infested wood, it can be beneficial in the initial tests of EDN or similar new fumigants, as research on these substances is extremely laborious and expensive. If the

substance tested does not show the desired activity in our simple model system, it will probably not work with naturally infested wood. It is believed that this validation method may help to develop new and effective EDN fumigation procedures and thereby provide certain contributions to the long-term protection of forests worldwide.

**Author Contributions:** O.D.: biological experiments, writing and editing; V.S.: conceptualization; M.M. and M.Z.: biological experiments; J.H.: EDN treatment and measurements. All authors have read and agreed to the published version of the manuscript.

**Funding:** This work was supported by the Technology Agency of the Czech Republic, Grant No. TH02030329.

**Conflicts of Interest:** The authors declare no conflict of interest. The funders had no role in the design of the study; in the collection, analyses, or interpretation of the data; in the writing of the manuscript; or in the decision to publish the results.

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
