# Peer review of "Inexpensive Screening Method to Validate the Efficacy of Ethanedinitrile Fumigant on the Forest Invasive Nematode Pest Bursaphelenchus xylophilus"

_sustainability, doi:10.3390/su12114765_

Round 1

Reviewer 1 Report

The study by Douda et al. is aimed at developing a quick and inexpensive methodology that can be used in lab conditions to test the efficacy of fumigants as treatment for the pine wood nematode, B. xylophilus, as well as other nematodes. In addition, the authors provide a concrete example of the application of this methodology testing the effect of ethanedinitrile on B. xylophilus. The topic is definitely of interest, as PWN is one of the most important forest pest occurring nowadays. The amount of data presented is really limited and I think that more data would be needed to understand whether the methodology is reliable or not; on the other hand, it has been submitted as brief report and this might balance the lack of further data. In any case, I have a number of major and minor comments that I hope can help the authors to improve this article.

MAJOR COMMENTS

1) Overview of methodologies already existing. The authors state that one of the main aim of their study is to develop an inexpensive method to test the efficacy of fumigants on PWN. What I think is completely missing is an overview of alternative methodologies commonly used to carry out this kind of test. The reader do not understand why the methodology proposed here is innovative and different than the methodologies already existing.

2) Authors report that one of the main aim of the study is “to validate the new method under different temperature and moisture conditions” (Line 67-68). However this was not done. Authors tested different EDN concentrations and different exposure times, but neither different temperatures nor different moisture conditions were considered. This represents a clear limitation. How did the authors decided which T° and which moisture to use? Is it possible that this can have affected PWN mortality?

3) Methodological details. Given that authors are describing a new methodology that can be used in the future by other people, more methodological details should be given (see, for example, specific comments below). Pictures or figures would certainly help. I really think that this would give more value to this article.

4) Statistical analysis. Despite it is quite irrelevant given that treatment led to 100% mortality, it is important to add more details about the statistical analysis used to analyze the data.

5) English must be improved. There are several sentences that are not written in the proper way and I recommend to ask the help of a mother tongue.

SPECIFIC COMMENTS

1) Line 32 and 34: citations for these statements should be included.

2) Line 32-42: I would somewhere mention that the PWN does not spread by itself but it requires its beetle vector, that in Europe is the pine sawyer beetle Monochamus galloprovincialis. This is one of the main reason why the eradication is almost impossible.

3) Line 23-44: “For this reason, all wood coming from affected countries as well as from North American countries with B. xylophilus has to be treated against this species of nematode [6].”. I suggest to modify this sentence as follows “For this reason, all wood imported from countries where B. xylophilus is native (i.e., North America) or has been introduced (i.e., Japan, China, Korea and Europe) must be treated to kill any nematode present within wood tissues”.

4) Line 52: add (EDN) after “ethanedinitrile”.

5) Line 52: I suggest to replace “EDN” with “The latter, in particular,”.

6) Line 55: add “In addition”, before “it decomposes”

7) Line 58-61: you should specify what you mean with “the bark beetle disaster”.

8) Line 63-64: “In particular, the latter factor affects the velocity of pest population development, metabolism and breathing”. This sentence must be improved. What do you mean with pest? Any pest? It might be better to say any insect or any invertabrate.

9) Line 76: “specimen” should be “specimens”

10) Line 73-74: where did you get the sawdust and how did you prepare it? More info are needed here.

11) Line 76: “1 ml contained from about 300 to 800 B. xylophilus specimen”. How did you get this data?

12) Line 76-79: “The average resulting moisture content of the treated sawdust was 39 %, this was established by weighting of the sawdust prepared for experiment, its drying (105 °C for 24 hours) and weighting of dry sawdust.”  I suggest to modify the sentence as follows: “The average resulting moisture content of the treated sawdust was 39 %; this value was obtained weighting the sawdust prepared for experiment, drying it out at 105 °C for 24 hours, and weighting dried sawdust.”

13) Line 82-84: “Five replicates were treated during each application.” This sentence has to be rewritten. Do you mean that five tubes were placed in the fumigation chamber for three times? If yes, this should be clearly stated and better explained as it is a crucial info.

14) Line 84-85: “The samples were withdrawn from the chamber during fumigation using a rubber septum in designated intervals”. I do not understand what the term “samples” refers to, and how “the samples were withdrawn”. More details should be given here.

15) Line 90: “Baermann’s funnel method”. Can you give some more details about this methodology?

16) Line 92-93: which statistical analysis did you use? More information are needed here. In addition, you should report values regarding statistical analysis in the results section and in tables.

17) Line 96-101: this part should be moved to the discussion section or deleted.

18) Line 109-111: “Even if it is impossible to perform research on the efficacy of ethanedinitrile against B. xylophilus in naturally infested logs under conditions found in the Czech Republic in a fumigation chamber,”. Why? Do you refer to the fact that the nematode is still not present?

19) Line 111-112: “it may be useful to evaluate the effect of fumigation under different temperature conditions”. Again, this is something that you did not test.

20) Line 116: “this experimental design was used for research on hydrogen cyanide”. Does this mean that the same methodology has been already described and applied in a previous study? Unclear what you want to say here.

Author Response

Thanks a lot for the expensive review of our work. We have accepted great major of your remarks; the manuscript was substantially improved by your review. Please see individual issues with our remarks in bold below:

English language and style

(x) Extensive editing of English language and style required 
( ) Moderate English changes required 
( ) English language and style are fine/minor spell check required 
( ) I don't feel qualified to judge about the English language and style 

Yes

Can be improved

Must be improved

Not applicable

Does the introduction provide sufficient background and include all relevant references?

( )

(x)

( )

( )

Is the research design appropriate?

( )

(x)

( )

( )

Are the methods adequately described?

( )

( )

(x)

( )

Are the results clearly presented?

( )

( )

(x)

( )

Are the conclusions supported by the results?

( )

(x)

( )

( )

Comments and Suggestions for Authors

The study by Douda et al. is aimed at developing a quick and inexpensive methodology that can be used in lab conditions to test the efficacy of fumigants as treatment for the pine wood nematode, B. xylophilus, as well as other nematodes. In addition, the authors provide a concrete example of the application of this methodology testing the effect of ethanedinitrile on B. xylophilus. The topic is definitely of interest, as PWN is one of the most important forest pest occurring nowadays. The amount of data presented is really limited and I think that more data would be needed to understand whether the methodology is reliable or not; on the other hand, it has been submitted as brief report and this might balance the lack of further data.

Yes, we are aware that data in our study are not sufficient for the full paper so we decided to submit it as short note.

In any case, I have a number of major and minor comments that I hope can help the authors to improve this article.

MAJOR COMMENTS

1) Overview of methodologies already existing. The authors state that one of the main aim of their study is to develop an inexpensive method to test the efficacy of fumigants on PWN. What I think is completely missing is an overview of alternative methodologies commonly used to carry out this kind of test. The reader do not understand why the methodology proposed here is innovative and different than the methodologies already existing.

More about methods of EDN testing against B. xylophilus used so far was inserted into Introduction. Advantages of method presented in our paper were also added.

2) Authors report that one of the main aim of the study is “to validate the new method under different temperature and moisture conditions” (Line 67-68). However this was not done. Authors tested different EDN concentrations and different exposure times, but neither different temperatures nor different moisture conditions were considered. This represents a clear limitation. How did the authors decided which T° and which moisture to use? Is it possible that this can have affected PWN mortality?

Corresponding sentence in Introduction was changed. Choice of temperature and moisture was based on our previous results with hydrogene cyanide; sentence on this was inserted into Methods.

Affection of PWN mortality by inappropriate moisture or temperature is possible however as nematodes in our control variants were living even after 40 hours we consider conditions of the experiment optimal.

3) Methodological details. Given that authors are describing a new methodology that can be used in the future by other people, more methodological details should be given (see, for example, specific comments below). Pictures or figures would certainly help. I really think that this would give more value to this article.

Two figures describing methods were included.

4) Statistical analysis. Despite it is quite irrelevant given that treatment led to 100% mortality, it is important to add more details about the statistical analysis used to analyze the data.

Using of basic descriptive statistics is usually not presented however we added sentence on this according to reviewer’s opinion.

5) English must be improved. There are several sentences that are not written in the proper way and I recommend to ask the help of a mother tongue.

This is bad surprise for us as the manuscript was edited by professional editing service before the first submission. We let the manuscript edit once again, please see the corresponding certificate as supplement file.

SPECIFIC COMMENTS

1) Line 32 and 34: citations for these statements should be included.

Accepted, 2 more citation included.

2) Line 32-42: I would somewhere mention that the PWN does not spread by itself but it requires its beetle vector, that in Europe is the pine sawyer beetle Monochamus galloprovincialis. This is one of the main reason why the eradication is almost impossible.

Accepted, remark on Monochamus sp. added.

3) Line 23-44: “For this reason, all wood coming from affected countries as well as from North American countries with B. xylophilus has to be treated against this species of nematode [6].”. I suggest to modify this sentence as follows “For this reason, all wood imported from countries where B. xylophilus is native (i.e., North America) or has been introduced (i.e., Japan, China, Korea and Europe) must be treated to kill any nematode present within wood tissues”.

Accepted and adjusted.

4) Line 52: add (EDN) after “ethanedinitrile”.

Accepted

5) Line 52: I suggest to replace “EDN” with “The latter, in particular,”.

Accepted

6) Line 55: add “In addition”, before “it decomposes”

Accepted

7) Line 58-61: you should specify what you mean with “the bark beetle disaster”.

Accepted, scientific name of the beetle added.

8) Line 63-64: “In particular, the latter factor affects the velocity of pest population development, metabolism and breathing”. This sentence must be improved. What do you mean with pest? Any pest? It might be better to say any insect or any invertabrate.

Accepted and adjusted.

9) Line 76: “specimen” should be “specimens”

Accepted

10) Line 73-74: where did you get the sawdust and how did you prepare it? More info are needed here.

Accepted, more information on this added.

11) Line 76: “1 ml contained from about 300 to 800 B. xylophilus specimen”. How did you get this data?

Accepted, more information on this added.

12) Line 76-79: “The average resulting moisture content of the treated sawdust was 39 %, this was established by weighting of the sawdust prepared for experiment, its drying (105 °C for 24 hours) and weighting of dry sawdust.”  I suggest to modify the sentence as follows: “The average resulting moisture content of the treated sawdust was 39 %; this value was obtained weighting the sawdust prepared for experiment, drying it out at 105 °C for 24 hours, and weighting dried sawdust.”

Accepted and changed.

13) Line 82-84: “Five replicates were treated during each application.” This sentence has to be rewritten. Do you mean that five tubes were placed in the fumigation chamber for three times? If yes, this should be clearly stated and better explained as it is a crucial info.

Accepted and adjusted.

14) Line 84-85: “The samples were withdrawn from the chamber during fumigation using a rubber septum in designated intervals”. I do not understand what the term “samples” refers to, and how “the samples were withdrawn”. More details should be given here.

Accepted and adjusted.

15) Line 90: “Baermann’s funnel method”. Can you give some more details about this methodology?

Accepted, details on method added.

16) Line 92-93: which statistical analysis did you use? More information are needed here.

Accepted and adjusted as described in point 4).

In addition, you should report values regarding statistical analysis in the results section and in tables.

Accepted, values were added in results.

17) Line 96-101: this part should be moved to the discussion section or deleted.

Accepted, the first sentence of results was incorporated into discussion.

18) Line 109-111: “Even if it is impossible to perform research on the efficacy of ethanedinitrile against B. xylophilus in naturally infested logs under conditions found in the Czech Republic in a fumigation chamber,”. Why? Do you refer to the fact that the nematode is still not present?

Yes, we are referring fact that B. xylophilus is not present in the Czech Republic, explanation was added.

19) Line 111-112: “it may be useful to evaluate the effect of fumigation under different temperature conditions”. Again, this is something that you did not test.

The sentence was reformulated.

20) Line 116: “this experimental design was used for research on hydrogen cyanide”. Does this mean that the same methodology has been already described and applied in a previous study? Unclear what you want to say here.

Yes, similar design was used for HCN earlier, purpose of this work was to evaluate if such design is applicable also for other fumigants (mainly promising EDN) with substantially different physical properties (EDN with very high density similar to methyl bromide vs. very light HCN and completely different distribution pattern in plant material and decomposition products).

Reviewer 2 Report

The manuscript provides practical use of a chemical for the control of a serious pathogen. Overall the paper is good, well written (despite some minor English corrections) and has sound scientific experiments. Specifics: L. 14: Please make species names (B. xylophilus) in italic; L. 16: “… methyl bromide has been..”; L. 35: reference and date? L. 40: as far as we know, no plant pathogen has been eradicated from the planet, so total eradication is impossible, or at least an illusion; L. 42: “… presents a threat…”; L. 43: “… in the entire continent”; L. 45: what about sulfuryl fluoride (SF)? L. 48: infrared  (IR) application seems to be a promising alternative; can you elaborate on this? L. 76: “… B. xylophilus individuals”.; L. 90: “… and the collected nematodes..”; L. 114: “… from aprox. 300 to 800…”; L. 136: “… cannot fully replace..”.

Author Response

Considering sulfuryl fluoride we are mentioning this substance in introduction, infrared treatment (like heat treatment) with its drawbacks is also mentioned in introduction. Otherwise we accepted all suggested changes, thank you very much.

English language and style

( ) Extensive editing of English language and style required 
(x) Moderate English changes required 
( ) English language and style are fine/minor spell check required 
( ) I don't feel qualified to judge about the English language and style 

Yes

Can be improved

Must be improved

Not applicable

Does the introduction provide sufficient background and include all relevant references?

( )

(x)

( )

( )

Is the research design appropriate?

(x)

( )

( )

( )

Are the methods adequately described?

(x)

( )

( )

( )

Are the results clearly presented?

(x)

( )

( )

( )

Are the conclusions supported by the results?

(x)

( )

( )

( )

Comments and Suggestions for Authors

The manuscript provides practical use of a chemical for the control of a serious pathogen. Overall the paper is good, well written (despite some minor English corrections) and has sound scientific experiments. Specifics: L. 14: Please make species names (B. xylophilus) in italic; L. 16: “… methyl bromide has been..”; L. 35: reference and date? L. 40: as far as we know, no plant pathogen has been eradicated from the planet, so total eradication is impossible, or at least an illusion; L. 42: “… presents a threat…”; L. 43: “… in the entire continent”; L. 45: what about sulfuryl fluoride (SF)? L. 48: infrared  (IR) application seems to be a promising alternative; can you elaborate on this? L. 76: “… B. xylophilus individuals”.; L. 90: “… and the collected nematodes..”; L. 114: “… from aprox. 300 to 800…”; L. 136: “… cannot fully replace..”.

Reviewer 3 Report

L14-L15: Bursaphelenchus xylophilus and B. xylophilus should be in italics

L36, L75: add Authotity and systematic classification to the scientific names when reported for the first time in the text

research design and methodology should be improved

Bibliography needs to be improve (Add more references i.e.: 

  • Thamarath Pranamornkith et al., 2014 Ethanedinitrile: Potential methyl bromide alternative to control Arhopalus ferus (Mulsant) in New Zealand sawn timber exports

Author Response

English language and style

( ) Extensive editing of English language and style required 
( ) Moderate English changes required 
(x) English language and style are fine/minor spell check required 
( ) I don't feel qualified to judge about the English language and style 

Yes

Can be improved

Must be improved

Not applicable

Does the introduction provide sufficient background and include all relevant references?

( )

(x)

( )

( )

Is the research design appropriate?

( )

(x)

( )

( )

Are the methods adequately described?

( )

(x)

( )

( )

Are the results clearly presented?

( )

(x)

( )

( )

Are the conclusions supported by the results?

( )

(x)

( )

( )

Comments and Suggestions for Authors

L14-L15: Bursaphelenchus xylophilus and B. xylophilus should be in italics

corrected

L36, L75: add Authotity and systematic classification to the scientific names when reported for the first time in the text

corrected

Please see our comments in bold:

Research design and methodology should be improved

Extensive adjustment of methodology and experimental design description was done.

Bibliography needs to be improve (Add more references i.e.: 

  • Thamarath Pranamornkith et al., 2014 Ethanedinitrile: Potential methyl bromide alternative to control Arhopalus ferus (Mulsant) in New Zealand sawn timber exports

Four more references including Thamarath Pranamornkith et al. were added.

Thank you for your review.

Round 2

Reviewer 1 Report

The paper has been greatly improved by the authors.